# SARS-CoV-2 Breakthrough Infections According to the Immune Response Elicited after mRNA Third Dose Vaccination in COVID-19-Naïve Hospital Personnel

**DOI:** 10.3390/biomedicines11051247

**Published:** 2023-04-23

**Authors:** Annapaola Santoro, Andrea Capri, Daniele Petrone, Francesca Colavita, Silvia Meschi, Giulia Matusali, Klizia Mizzoni, Stefania Notari, Chiara Agrati, Delia Goletti, Patrizio Pezzotti, Vincenzo Puro

**Affiliations:** 1National Institute for Infectious Diseases, Lazzaro Spallanzani IRCCS, 00149 Rome, Italy; 2Bambino Gesù Children’s Hospital, IRCCS, 00165 Rome, Italy; 3Department of Infectious Diseases, National Institute of Health (ISS), 00161 Rome, Italy

**Keywords:** SARS-CoV-2, mRNA vaccine against SARS-CoV-2, COVID-19, breakthrough infections, immune response

## Abstract

Background: Vaccine-induced SARS-CoV-2-anti-spike antibody (anti-S/RBD) titers are often used as a marker of immune protection and to anticipate the risk of breakthrough infections, although no clear cut-off is available. We describe the incidence of SARS-CoV-2 vaccine breakthrough infections in COVID-19-free personnel of our hospital, according to B- and T-cell immune response elicited one month after mRNA third dose vaccination. Methods: The study included 487 individuals for whom data on anti-S/RBD were available. Neutralizing antibody titers (nAbsT) against the ancestral Whuan SARS-CoV-2, and the BA.1 Omicron variant, and SARS-CoV-2 T-cell specific response were measured in subsets of 197 (40.5%), 159 (32.6%), and 127 (26.1%) individuals, respectively. Results: On a total of 92,063 days of observation, 204 participants (42%) had SARS-CoV-2 infection. No significant differences in the probability of SARS-CoV-2 infection for different levels of anti-S/RBD, nAbsT, Omicron nAbsT, or SARS-CoV-2 T cell specific response, and no protective thresholds for infection were found. Conclusions: Routine testing for vaccine-induced humoral immune response to SARS-CoV-2 is not recommended if measured as parameters of ‘protective immunity’ from SARS-CoV-2 after vaccination. Whether these findings apply to new Omicron-specific bivalent vaccines is going to be evaluated.

## 1. Introduction

Between the first cases recognized in December 2019 and March 2023, three years after the declaration of the SARS-CoV-2 pandemic, the WHO reported that around 800 million COVID-19 cases have been confirmed worldwide, with 7 million deaths. These findings do not take into account the large number of undiagnosed or unreported infections, thus likely representing a strong underestimation of the global infection burden [1,2,3]. 

Starting in November 2021, the SARS-CoV-2 pandemic has been characterized, worldwide, by a massive surge in infections driven by the Omicron variant of the virus (B.1.1.529), and its sublineages, owing to its higher transmissibility and immune-escaping profile [4,5].

Just after one year from the identification of the new coronavirus, several vaccines became available and a worldwide mass vaccination campaign was conducted in 2021 and 2022, covering around 70% of the world’s population as of 5 April 2023 [6].

Consolidated evidence has been accumulated on the effectiveness of COVID-19 vaccines in preventing SARS-CoV-2 infection in the short term and, more importantly, in reducing the risk of severe COVID-19 and death overall. Higher effectiveness can be achieved, in particular, when booster doses are provided to overcome the waning of immunity that occurs after the first few months following vaccination [7,8,9]. Consolidated evidence is also available on the general safety of COVID-19 vaccines. Severe adverse events, such as anaphylaxis, myocarditis and pericarditis, have been reported; although they appear to be extremely rare; i.e., a few cases per million doses [10,11,12]. The WHO’s Strategic Advisory Group of Experts on Immunization (SAGE) were still emphasizing the clear benefit of vaccination for individuals at higher risk of severe COVID-19 complications as of 28 March 2023 [13].

COVID-19 is characterized by a wide range of clinical pictures from asymptomatic cases or mild flu-like syndrome, to pneumonia, severe acute respiratory infection (SARI) or acute respiratory distress syndrome (ARDS), and death [14,15,16]. Moreover, about 10-20% of patients who recovered from COVID-19 are sufferers of prolonged clinical sequelae, defined as long-COVID-19 [17,18]. Some evidence is accumulating on an intrinsic milder clinical severity of Omicron, in comparison to previous variants, which largely contributed to the wide level of immunity reached in the population after natural infection and/or vaccination and strongly modified the pandemic’s features [19]. 

Indeed, although the Omicron variant has been associated with reduced effectiveness against infection of first-generation COVID-19 vaccines based on the spike protein of the ancestral Wuhan lineage, it has been demonstrated that first-generation mRNA monovalent COVID-19 vaccines are still effective in preventing severe outcomes from infection, and a three-dose mRNA vaccination also increased immunity against Omicron [8,9,20,21,22].

Vaccines induce both humoral immunity based on antibodies binding the SARS-CoV-2 spike protein that neutralize the virus, and cellular immunity including virus-specific B- and T-cells providing long-term memory and promptly expanding following re-exposure to antigens [23,24]. 

SARS-CoV-2-anti-spike antibody titers, which correlate with the presence of neutralizing antibodies, are often used as a marker of immune protection and to anticipate the risk of breakthrough infections. However, no clear cut-off indicating protective immunity against infection or severe outcomes, i.e., an immune marker that is correlated with vaccine efficacy, is available [24,25,26,27,28]. These cut-offs would be important to find correlates of protection, especially for vulnerable populations with immune impairment or on treatment with immune-suppressive drugs [29,30,31,32]. 

In this study we assessed the incidence of SARS-CoV-2 vaccine breakthrough infections in COVID-19-free personnel, after receiving the mRNA third dose vaccination. The aim was to evaluate whether the peak levels of B- and T-cell immune response elicited one month after receiving the booster, influenced the risk of subsequent infection, as well as whether protective thresholds could be identified. 

## 2. Materials and Methods

### 2.1. Setting and Patient Selection

On 27 December 2020, based on Italian Ministry of Health recommendations, the National Institute for Infectious Diseases (INMI) L. Spallanzani started a vaccination campaign against SARS-CoV-2, targeted to its staff. The campaign started with clinical and laboratory workers, and then opened to all its personnel. The BNT162b2 mRNA-based monovalent vaccine [Comirnaty/BioNTech (Cambridge, MA, USA) Pfizer (New York, NY, USA) 30 μg] was the only one available at that time. 

On 15 October 2021, a booster dose campaign started: the BNT162b2 Comirnaty or mRNA-1273 (Spikevax, Moderna 50 μg) vaccine was offered. Since the beginning of the vaccination campaign, voluntary surveillance was implemented to follow up both the humoral and cell-mediated response to the vaccine [33]. The proposal was approved by the Ethics Committee of the National Institute for Infectious Diseases Lazzaro Spallanzani (Approval n. 297 and n. 442, 2020/2021).

Based on the surveillance protocol, following written informed consent, blood samples were collected one month after the third dose. For the present study, staff who had completed the primary vaccine schedule, who received the booster dose before 30 December 2021, and consented to post-booster evaluation of immune response, were eligible for inclusion. Subjects with a previous SARS-CoV-2 diagnosis, i.e., positive to the antigenic and/or molecular test on the swab sample by real-time polymerase chain reaction (PCR), positive to anti-N, or with an abrupt increase in SARS-CoV-2 anti-S/RBD levels not related to vaccination, were excluded (Box Figure 1).

We evaluated in all participants the IgG antibody levels against the nucleocapsid protein (anti-N) and against the spike receptor binding domain (anti-S/RBD) of SARS-CoV-2. In a smaller sample of unselected participants we assessed both the neutralizing antibody titers (nAbsT) against the ancestral Whuan SARS-CoV-2 (W-D614G) and against the BA.1 Omicron variant, and T-cell-specific response by the IFN-γ detection. 

In our Institute, based on the internal occupational health protocol; the staff were tested for SARS-CoV-2 through nasal or nasopharyngeal swab (rapid antigen and/or real-time polymerase chain reaction, as appropriate); in cases of occupational or community exposure to a person diagnosed or suspected of COVID-19, in cases of symptoms suggestive of SARS-CoV-2 infection, or periodically on a voluntary basis.

To evaluate the risk of vaccine breakthrough infection, individuals were followed until 30 June 2022. This study period coincided with two concentrated and intense waves of SARS-CoV-2 infections, in the Latium region and in the country, characterized by the progressive ongoing replacement of the Delta variant of the virus with the Omicron variant: the former from December to February (peaking on 14 January with 15,000 cases in the region), and the latter from March to May (peaking on 27 March with 11,000 cases) [34]. We defined a booster vaccination breakthrough case as a rapid antigen and/or real-time polymerase chain reaction positive nasal or nasopharyngeal sample obtained more than 5 days after receiving a third vaccine dose. 

### 2.2. Laboratory

We used two commercial chemiluminescence microparticle antibody assays (CMIA), the SARS-CoV-2 specific anti-N and the anti-S/RBD tests [AdviseDx SARS-CoV-2 IgG II and SARS-CoV-2 IgG II Quant, respectively, ARCHITECT^®^ (Chicago, IL, USA) i2000sr Abbott Diagnostics, Chicago, IL, USA] according to the manufacturer’s instruction; index > 1.4 and binding antibody units (BAU)/mL > 7.1 are considered positive, respectively.

We measured the neutralizing antibody titers (nAbsT) by micro-neutralization assay based on live SARS-CoV-2 virus for W-D614G (Ref-SKU: 008V–04005, from EVAg portal), and BA.1 (GISAID accession ID EPI_ISL_7716384). We inactivated the serum samples at 56 °C for 30 min and titrated in duplicate in 7 twofold serial dilutions (starting dilution 1:10). Each serum dilution (50 μL) and medium (50 μL) containing 100 TCID50 SARS-CoV-2 were mixed and incubated at 37 °C for 30 min. We defined as neutralizing the highest serum dilution inhibiting at least 90% of the cytopathic effect on Vero E6 cells, and neutralizing antibodies (nAbs) were categorized as undetectable if titers were <1:10 [35].

We evaluated the SARS-CoV-2-specific T-cell response functions, bu by collecting the peripheral blood in heparin tubes. Whole blood was then stimulated or not with a pool of peptides spanning the spike protein (Miltenyi Biotech, Bergisch Gladbach Germany) at 37 °C (5% CO_2_). We employed a superantigen (SEB) as the positive control. After 16–20 h of stimulation we harvested plasma which was then stored at −80 °C. We assessed the T-cell-specific response through the quantification of the IFN-γ released in plasma by an automated ELISA (ELLA, Protein Simple). The IFN-γ detection limit of these assays was 0.17 pg/mL [36]. 

### 2.3. Statistical Analysis

Descriptive statistics such as absolute and relative percentage for categorical variables, and mean, median, standard deviations, range, and interquartile ranges were calculated to summarize and to visualize the data. Data on different types of immune response were transformed and analyzed on a logarithmic scale. 

Kaplan–Meier curves were used to estimate the probability of being diagnosed with SARS-CoV-2 infection at different times following the immune response measurement at around one month after booster dose. The measurements were categorized into four groups using as cut-off value the median and the quartiles of each distribution. During preliminary analyses several other cut-off values were considered, but results remained substantially unchanged.

Cox proportional hazards regression multivariable models were performed to estimate hazard ratios of being diagnosed with SARS-CoV-2 infection for the levels of any type of measured response; adjusting simultaneously for the characteristics of the individual; (i.e., sex, age at first dose vaccination, type of vaccine used as booster dose (Comirnaty or Spikevax), type of work (direct/not direct contact with patients)). The final model was chosen using backward criterion. A *p*-value less than 0.05 was considered statistically significant.

## 3. Results

A total of 754 COVID-naïve individuals received the third dose of mRNA-vaccine against SARS-CoV-2 (Figure 1). The study included 487 (65%) individuals for whom data on anti-/SRBD were available after the booster dose. Participants had received the first dose between 27 December 2020 and 11 June 2021, with more than 75% vaccinated within January 2021. The second dose was administered after 21 and 28 days for those receiving Comirnaty or Spikevax, respectively. The booster dose was administered between 15 October and 30 December 2021. Samples at around one month after booster dose were taken between 3 November 2021 and 15 February 2022 (Figure 1).

Up to 30 June 2022, on a total of 92,063 days of observations after the sample taken at around one month after the booster dose, 204 participants (41.9%) were diagnosed with SARS-CoV-2 infection. Clinically, all but one case had no or mild symptoms without pneumonia; one necessitated hospitalization due to moderate bilateral interstitial pneumonia and recovered after 10 days.

Table 1 summarizes the main characteristics of the study subjects, according to breakthrough infections. More than 75% of the participants were women, as well as participants having direct contacts with patients; overall, the median age was 45 years (range 23–66). 

Overall, the anti-S/RBD geometric mean one month after the booster dose was 3521 BAU (IQR: 2233–5493). Table 2 shows the estimated anti-S/RBD mean values one month after the third dose, with 95% confidence intervals (95% CI) stratified by age and sex.

Table 3 shows the estimated adjusted hazard ratios of different levels of anti-S/RBD at one month after mRNA booster dose obtained from different models, adjusting simultaneously for other characteristics of the participants. In all these models age was inversely associated with the risk of being diagnosed with SARS-CoV-2 infection. Anti-S/RBD between 2500 and 4000 BAU/mL had a hazard ratio significantly higher (*p* < 0.01) than that of participants with <2500 BAU/mL; in an analysis where anti-S/RBD were grouped below and above 4000 BAU/mL, hazard ratios (in each model performed) were not significantly different. 

In subsets of 197 (40.5%), 159 (32.6%), and 127 (26.1%), participants’ nAbsT, Omicron-nAbsT, and SARS-CoV-2–specific T-cell response evaluated by the detection of IFN-γ production, respectively, were measured one month after the booster dose. Table 1 shows the geometric mean and, also, the interquartile range for these parameters, stratified by breakthrough infections or not at the end of follow-up. No multivariable models were performed given the low number of participants who were sampled. 

Figure 2 shows the cumulative probabilities (estimated by the Kaplan–Meier method) of being diagnosed with SARS-CoV-2 breakthrough infection by days following the anti-S/RBD measurements at one month after booster dose, stratified by different levels of response. Participants with higher anti-S/RBD values exhibited similar curves to those of participants with lower values (Figure 2A). Although some graphical differences between cumulative Kaplan–Meier curves regarding the overall nAbsT and those against Omicron variant can be observed, these were not statistically significant (Figure 2B and Figure 2C, respectively). Similarly, no significant differences in the cumulative probability of being diagnosed with SARS-CoV-2 infection were found for different levels of T-cell-specific response evaluated by the detection of IFN-γ in response to SARS-CoV-2 antigens (Figure 2D). 

Regarding the individual characteristics of the participants, we found a significantly higher cumulative probability of being diagnosed with SARS-CoV-2 infection for subjects aged ≤45 compared to those >45 years old (*p* < 0.001); no significant differences were found by type of mRNA vaccine administered, sex, and direct/not direct contact with patients (Figure 3).

## 4. Discussion

This study showed that in our cohort of health personnel naive for COVID-19 infection and vaccinated with three doses of mRNA SARS-CoV-2 vaccine, a total of 204 participants (41.9%) developed a breakthrough infection within 22 weeks after booster vaccination. The breakthrough infection was independent of the levels of humoral or T-cell-specific response elicited one month after the mRNA vaccination booster dose.

Anti-S/RBD response one month after the mRNA vaccination booster dose was robust and also had a good correlation with overall and Omicron-specific nAbsT; conversely, the correlation was low between the SARS-CoV-2-specific T-cell response and the nAbs T responses. 

The age and sex of the participants were significantly associated with the anti-S/RBD response at different times; however, the magnitude of this effect was substantially low.

In the subgroup of workers for whom the nAbsT was performed, a good neutralizing activity against the ancestral SARS-CoV-2 strain was observed. However, no difference was found between those who had SARS-CoV-2 breakthrough infections during the study period and those who did not. These findings confirm that the Omicron variant spike protein can escape from neutralizing antibodies elicited in recipients of three monovalent mRNA vaccine doses.

Furthermore, an anti-Omicron-neutralizing response was observed in the smaller subgroup of workers for whom the test was performed, with titers, not unexpectedly, lower against Omicron compared with those against the ancestral strain. Despite anti-Omicron geometric mean titers being slightly higher in non-breakthrough workers, the difference was not statistically significant. Similar findings have been observed about T-cell-specific response in the subset of participants tested. 

Our study is consistent with previous findings showing that booster immunization with first-generation mRNA monovalent vaccines could still improve the immune response against the Omicron variant; although this increased specific immunity did not associate with prolonged protection from infection; likely due to its waning and to Omicron’s capacity to escape vaccine-elicited antibodies [9,37,38,39]. 

Thus, in our study, none of the humoral and T-cell immune response parameters considered after one month from booster, were associated with the cumulative probability of being diagnosed with SARS-CoV-2 infection during the follow-up period.

This was also confirmed in multivariable analysis of the anti-S/RBD response, adjusting for the characteristics of the participants. The only characteristic associated with the cumulative probability of breakthrough infection was age, with those >45 years old having a lower probability, as generally observed [3]. Very likely, this age group effect is the result of a different degree of socialization and other behavioral risk factors in the community, possibly greater among younger people.

Our findings are consistent with those found in several other studies [24,40,41,42], but conflict with others [12,43,44,45].

Similar findings were indeed found after the two-dose primary mRNA vaccination, as well as vaccines containing the inactivated ancestral strain of SARS-CoV-2, although in smaller series and before Omicron diffusion [40,41,42].

In contrast to our observed features, in a study conducted in Israel, a lower humoral response after the second dose of the BNT162b2 mRNA vaccine was associated with a higher risk of infection, mostly due to the B.1.1.7 (alpha) variant of the virus [43]. 

In other studies, potential threshold values of anti-S-specific antibodies elicited post-vaccination were associated with a higher probability of developing breakthrough infections; however, these values were widely different among the different reports [28,44,45]. 

In almost all these studies, a high antibody titer does not guarantee absolute protection towards mild and asymptomatic breakthrough infections.

Our study presents some limitations. First, it was conducted in a single center where only first-generation monovalent mRNA vaccines were used; and healthy, young, and middle-aged adults, predominantly female, without previous SARS-CoV-2 infection; were recruited. Moreover, the study population is a convenience sample of staff who consented to post-booster evaluation, although unselected. Thus, the observed findings should not be generalized to different type of vaccines or schedules and to a different population, especially in those patients who are immunocompromized and/or treated with immune-suppressive therapies [29,30,31,32].

In particular, whether and which immune response level peaked after booster dose translates to the risk of COVID-19 severe outcomes, needs further investigation. A relationship between neutralization level after SARS-CoV-2 vaccination and protection against severe COVID-19 has been demonstrated, and the level of neutralizing antibodies associated with protection against severe disease was much lower than the level required to provide protection against infection [25]. 

Another possible source of bias in our results is the presence of undocumented infections among the “uninfected” group of individuals. We are confident that the high rate of testing after known potential exposure, and symptoms development in our personnel, reduced this potential bias. 

In addition, another limitation could be the effect of different speeds of immune response decline; that was not evaluated: lower levels may have been reached faster among participants who developed breakthrough infection regardless of the peak level elicited after the third dose; however, no risk factors for an accelerated waning were recognized in the study population [44,45].

Finally, we did not perform genotyping of virus variants and relied on national and regional surveillance data to support the claim that the viral epidemiology in our area followed national trends, with the Omicron variant taking over after December 2021 [34]. 

## 5. Conclusions

In conclusion, in our study population, the level of specific humoral and cellular immune response elicited at one month from the first-generation monovalent mRNA vaccine booster was not associated with the cumulative probability of being diagnosed with SARS-CoV-2 infection, and no protective threshold for infection was stated. 

The follow-up period was characterized by the increasing spread of the immune-evasive Omicron variant of SARS-CoV-2, and likely associated with the waning of post-booster immune response.

These findings are consistent with current recommendations stating that antibody testing after COVID-19 vaccination should not be used to measure ‘protective immunity’ for SARS-CoV-2 infection after vaccination in immune-competent individuals [46]. The role of testing for vaccine-induced humoral immune responses to SARS-CoV-2 remains of limited clinical value. Several studies have demonstrated that populations with a compromised humoral immunogenicity such as transplant patients, people living with advanced/uncontrolled HIV infection, rheumatoid arthritis, or multiple sclerosis, may be non- or low responders to the SARS-CoV-2 vaccine and at risk of more severe COVID-19 outcomes [29,30,31,32]. In these vulnerable populations there could be the need to identify those patients who would benefit from pre-exposure prophylaxis with anti-SARS-CoV-2 monoclonal antibodies, when appropriate according to susceptibility to predominant variant [16,47].

Accurate correlates of immune protection, to discriminate the risk of infection at an individual level, are still needed and further research in this field is necessary. Whether these findings apply to vaccination or boosters with the new Omicron-specific bivalent vaccines, as well as to natural or hybrid immunity that may offer higher and more durable protection, is going to be evaluated.

Currently, the COVID-19 pandemic is characterized by SARS-CoV-2 Omicron variant sublineages still evolving and spreading in a population with a high level of immunity because of vaccination or previous COVID-19 infection, or both [1,2,4]. In this scenario, as recently stated by the WHO’s Strategic Advisory Group of Experts on Immunization [13], it is extremely important to continue to offer vaccination to people at higher risk of developing severe disease from SARS-CoV-2 infection. Older adults and those with underlying vulnerable conditions should be efficiently targeted to receive the primary vaccine series as well as additional boosters. In addition, efforts should be made to continue the control of vaccination at a public health level in order to protect these high-priority population groups. 

## Figures and Tables

**Figure 1 biomedicines-11-01247-f001:**
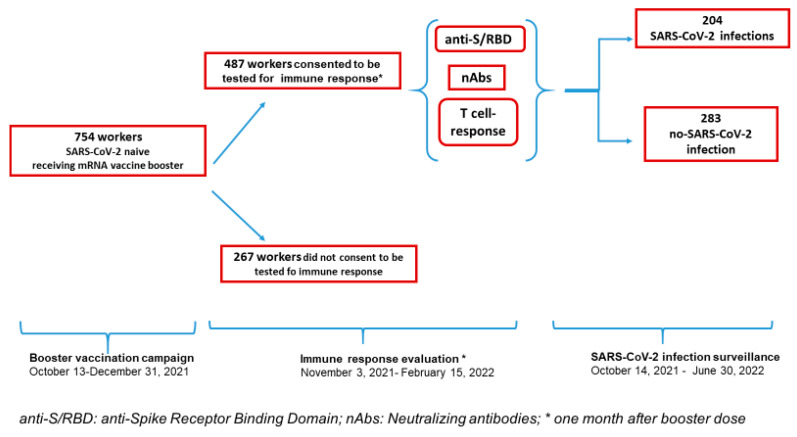
Setting and Patient Selection.

**Figure 2 biomedicines-11-01247-f002:**
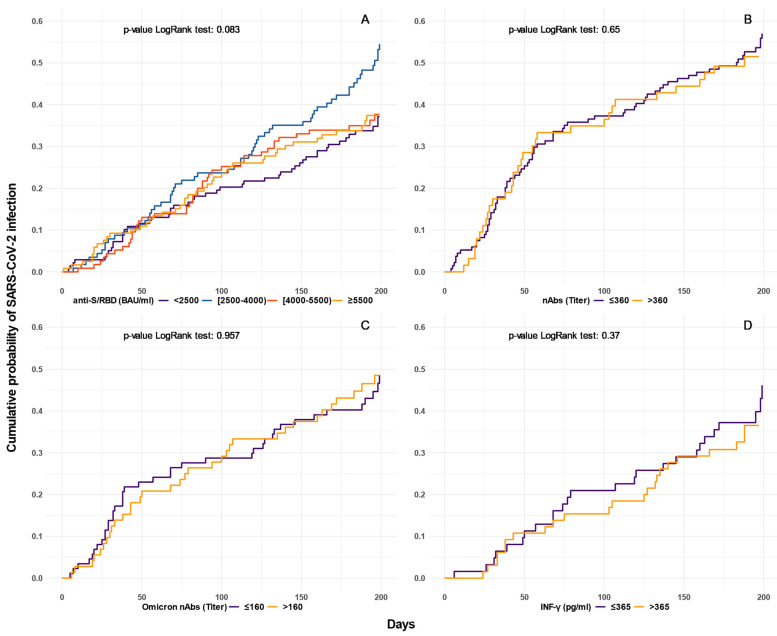
Cumulative probability of being diagnosed with SARS-CoV-2 infection following mRNA vaccine booster dose according to anti-S/RBD, nAbs, Omicron nAbs, IFN-γ response (Panel (**A**–**D**), respectively); anti-S/RBD: anti-Spike Receptor Binding Domain; BAU: binding antibody units; nAbs: Neutralizing antibodies; IFN: Interferon; pg:picograms. The levels are categorized using as cut-off value the median and the quartiles of each distribution. Kaplan–Meier curves estimate the probability of SARS-CoV-2 infection at different times.

**Figure 3 biomedicines-11-01247-f003:**
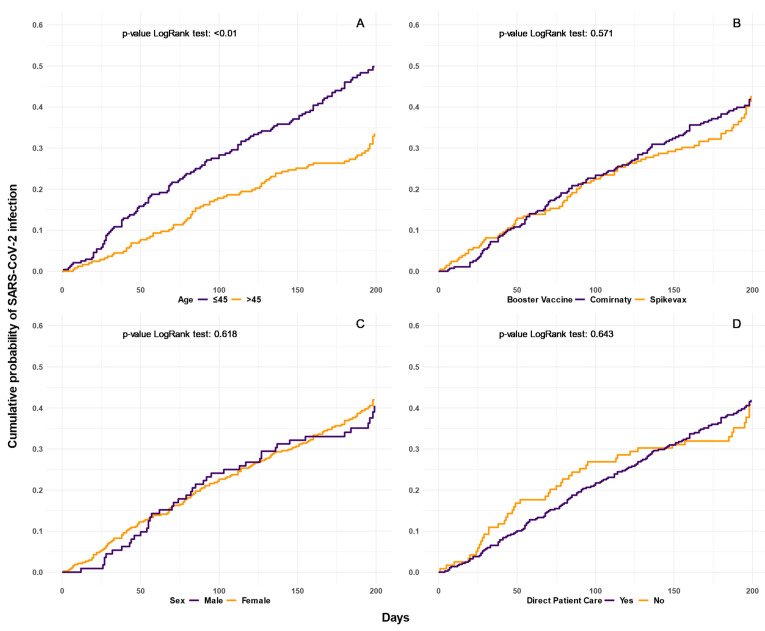
Cumulative probability of being diagnosed with SARS-CoV-2 infection at one month after mRNA vaccine booster dose by age, type of mRNA vaccine booster administered, sex, and direct/not direct contact with patients (Panel (**A**–**D**), respectively). Kaplan–Meier curves estimate the probability of SARS-CoV-2 infection at different times.

**Table 1 biomedicines-11-01247-t001:** Characteristics of the participants.

Variable		SARS-CoV-2 Infected		Not Infected		Total	*p*-Value **
Sex	Female	158	42.1%	217	57.9%	375	
	Male	46	41.1%	66	58.9%	112	0.93
Direct patient care	No	47	39.5%	72	60.5%	119	
	Yes	157	42.7%	211	57.3%	368	0.62
Booster dose vaccine	Spikevax	79	37.8%	130	62.2%	209	
	Comirnaty	125	45.0%	153	55.0%	278	0.14
Age at first dose vaccination (median, IQR)		44	(32–50)	48	(37–54)	487	0.001
Days from booster dose to sample (median, IQR)		32	(30–35)	32	(30–34)	487	0.98
anti-S/RBD * (BAU/mL geometric mean, IQR)		3752	(2380–5318)	4070	(2176–5662)	486	0.49
nAbs titer * (geometric mean, IQR)		320	(160–640)	320	(160–640)	197	0.40
Omicron nAbs titer * (geometric mean, IQR)		80	(40–160)	80	(80–160)	159	0.58
T-cell specific response measured by IFN-γ * (pg/mL; geometric mean, IQR)		321	(170–688)	442	(198–852)	127	0.21
Total		204	41.9%	283	58.1%	487	

IQR: Inter Quartile Range; BAU: binding antibody units; anti-S/RBD: anti-Spike Receptor Binding Domain; nAbs: Neutralizing antibodies; IFN: Interferon; * one month after booster dose; ** Chi square test was used to compare categorical variables; U Mann-Whitney test was used for continuous variables.

**Table 2 biomedicines-11-01247-t002:** Estimated anti-S/RBD through a multivariable linear regression mixed model, according to sex and age.

Times	Sex	Age	Estimated Geometric Mean BAU/mL	95% CI	
One month after booster dose	Male	≤45 years	3843	3285	4497
	Male	>45 years	3353	2883	3899
	Female	≤45 years	3752	3408	4130
	Female	>45 years	3273	2967	3609

Note: Direct patient care (yes/no) and exact time between booster dose administration and date of sample were not included in the final model because these did not improve the goodness of fit to raw data; BAU: binding antibody units; anti-S/RBD: anti-Spike Receptor Binding Domain CI: Confidence intervals.

**Table 3 biomedicines-11-01247-t003:** Estimated hazard ratios of SARS-CoV-2 breakthrough infection according to anti-S/RBD obtained from four Cox proportional hazards regression multivariable models.

		Model 1			Model 2			Model 3			Model 4		
		HR	95% CI	*p*-Value	HR	95% CI	*p*-Value	HR	95% CI	*p*-Value	HR	95% CI	*p*-Value
anti-S/RBD (BAU/mL)	<2500	reference			reference			reference			reference		
	2500–4000	1.45	1.00–2.10	0.05	1.45	1.00–2.10	0.05	1.44	1.00–2.10	0.05	1.45	1.00–2.10	0.05
	4001–5500	1.05	0.70–1.57	0.81	1.05	0.70–1.57	0.81	1.05	0.70–1.56	0.82	1.04	0.70–1.55	0.84
	>5500	1.01	0.67–1.51	0.98	1.01	0.67–1.51	0.98	1.00	0.67–1.51	0.99	0.99	0.67–1.48	0.97
Age	≤45	reference			reference			reference			reference		
	>45	0.59	0.44–0.78	<0.01	0.59	0.44–0.78	<0.01	0.59	0.45–0.78	<0.01	0.59	0.45–0.78	<0.01
Vaccine used as booster dose	Comirnaty	reference			reference			reference			-	-	-
	Spikevax	0.96	0.71–1.28	0.77	0.96	0.72–1.28	0.77	0.96	0.72–1.28	0.77	-	-	-
Direct patient care	Yes	reference			reference			-	-	-	-	-	-
	No	1.03	0.74–1.44	0.86	1.03	0.74–1.44	0.86	-	-	-	-	-	-
Sex	Male	reference			-	-	-	-	-	-	-	-	-
	Female	0.99	0.71–1.39	0.97	-	-	-	-	-	-	-	-	-

Note: Model 1 included all independent variables; from model 2 to model 4, non-significant variables were excluded by log-likelihood ratio test (*p* > 0.05) using a backward criterion. HR: Hazard ratio; CI: Confidence intervals; anti-S/RBD: anti-Spike Receptor Binding Domain; BAU: binding antibody units.

## Data Availability

The raw data supporting the conclusions of this article will be made available by the authors, without undue reservation.

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
