# Peer review of "SARS-CoV-2 Breakthrough Infections According to the Immune Response Elicited after mRNA Third Dose Vaccination in COVID-19-Naïve Hospital Personnel"

_biomedicines, 2023, doi:10.3390/biomedicines11051247_

Round 1

Reviewer 1 Report

The article by Annapaola Santoro et al. studies the incidence of COVID-19 infection, emphasizing the omicron strain, due to the advancement of the SARS-CoV-2 vaccine in personnel at a hospital in Italy. In addition, a relationship is made with the B and T cell immune response after the third dose of mRNA vaccination. The work is clear and relevant to her field. The conclusion is supported by the results. Therefore, it is suggested to accept the manucript after attending to the following minor recommendations:

-       Include how many patients were considered for the study from the protocol.

-       Lines 97-99. Use square brackets when double parentheses. For example:

[AdviseDx SARS-CoV-2 IgG II and SARSCoV-2 IgG II Quant, respectively, ARCHITECT® (Chicago, IL, USA) i2000sr Abbott  Diagnostics,Chicago, IL, USA].

-       Table 1. Define IQR

-       Table2. Define BAU

-       Table 3. Define BAU

-       Figure 1. Place the caption below the illustration.

-       Line 25. Change "on the probability…" to "in the probability..."

-       Line 64. Change "to vaccine" to "to the vaccine"

-       Line 81: Change "staff is..." to " the staff is..."

-       Line 136: Change "second" to " the second"

-       Line 271: Add a comma after the word "studies"

-       Line 295: Add a comma after the word "population"

-       Line 296: Change "from original" to "from the original"

Reviewer 2 Report

1)  Abstract Methods: The study included 487 19 individuals for whom data on anti-S/RBD were available. Neutralizing antibody titers (nAbsT) 20 against the original Whuan SARS-CoV-2, and the BA.1 Omicron variant, and SARS-CoV-2 T-cell 21 specific response were measured in subsets of 197 (40.5%), 159 (32.6%), and 127 (26.1%) individuals, 22 respectively. Results: On a total of 92,063 days of observation, 204 participants (42%) had SARS- 23 CoV-2 infection. Participants with higher anti-S/RBD values had similar incidence curves to those 24 with lower values; no protective threshold was found. No significant differences on the probability 25 of SARS-CoV-2 infection was found for different levels of nAbsT, Omicron nAbsT, or SARS-CoV-2 26 T cell specific response. Please insert the most important statistically significant values to support the data.

2) 1. Introduction L34-36. Since November, 2021, the SARS-CoV-2 pandemic has been characterized, worldwide, by  a massive surge of infections driven by the Omicron variant of the virus (B.1.1.529), and  its sublineages, owing to its higher transmissibility and immune escaping profile [1,2]. Please add some information regarding SARS-CoV-2 pandemic. The Authors should cite some more of the recent studies in the area on this topic and discuss in the Discussion Section how their study adds to the current data:

a-  COVID-19 and Post-Acute COVID-19 Syndrome: From Pathophysiology to Novel Translational Applications. Biomedicines 202210, 47. https://doi.org/10.3390/biomedicines10010047

b- Different Methods to Improve the Monitoring of Noninvasive Respiratory Support of Patients with Severe Pneumonia/ARDS Due to COVID-19: An Update. J Clin Med. 2022 Mar 19;11(6):1704. doi: 10.3390/jcm11061704.

3) Introduction. L52-53 The aim of this study was to describe the incidence of SARS-CoV-2 vaccine breakthrough  infections in COVID-19 free personnel of our hospital, according to the B and T-cell  immune response elicited after mRNA third dose vaccination. Please improve the description of study aim and underline the novelty of the study.

4) Statistical analysis. Please add the statisticaly significant p value.

5) 3. Results L133-145 The study included 487 individuals for whom data on anti-/SRBD were available after the booster dose. Participants received the first dose between December 27th 2020 and June 11th  2021 with more than 75% vaccinated within January 2021. Underline the statistically significant p values to support the data.

6) Table 1: Characteristics of the participants. Please, specify if there are differences between the two groups.

7) Table 3: Estimated hazard ratios of SARS-CoV-2 breaktrough infection according to anti-S/RBD obtained from four Cox 171 proportional hazards regression multivariable models. Please check the table, a part was cut

8) 5. Conclusions L294-298. In conclusion, in our study population the level of specific humoral and cellular immune  response elicited after one month from original mRNA vaccine booster was not associated  with the cumulative probability of being diagnosed with SARS-CoV-2 Omicron infection  during the follow-up period, alongside the likely waning of immune response, and no  protective threshold has been stated. Could you please improve this paragraph? Please underline the possible clinical implications of the study.

Reviewer 3 Report

In this study, the authors aimed to describe the incidence of SARS-CoV-2 vaccine breakthrough infections in COVID-19 free personnel from a hospital, according to the B and T-cell immune response elicited after mRNA third dose vaccination.

Comments and suggestions

A summarized scheme with all the steps included in this study is useful for readers for a better understanding of the MS.

Introduction section:

Lines 35-37: add more data about the difference between the two variants of SARS-CoV-2

Line 38: mention which is the first  generation of COVID-19 vaccines against infection

Line 41: “three-dose vaccination increased immunity also against Omicron” – mention the type of vaccine used

Lines 43-45: add more data regarding the immunity developed after the administration of mRNA vaccines. Also, mention some adverse effects and disadvantages.

Material and methods: highlight more inclusion criteria of the patients in the study. Exclusion criteria also missing. Ethical approval for this study is missing.

Figure 1: the graphs are not legible and it is not very clear what is represented. Revise at a higher resolution, with legible writing and add more explanations in the legend regarding the content.

Mention more limitations of this study

What new perspectives for human health does this MS have?

Consider revision accordingly.

Round 2

Reviewer 3 Report

No answer given.

Author Response

Thank you for all the suggestions

Vincenzo Puro